# Circulating Tumor DNA Is a Variant of Liquid Biopsy with Predictive and Prognostic Clinical Value in Breast Cancer Patients

**DOI:** 10.3390/ijms242317073

**Published:** 2023-12-02

**Authors:** Tatiana M. Zavarykina, Polina K. Lomskova, Irina V. Pronina, Svetlana V. Khokhlova, Marina B. Stenina, Gennady T. Sukhikh

**Affiliations:** 1N.M. Emanuel Institute of Biochemical Physics of the Russian Academy of Sciences, Moscow 119334, Russia; brenner123@mail.ru; 2“B.I. Kulakov National Medical Research Center of Obstetrics, Gynecology, and Perinatology of Ministry of Health of the Russian Federation, Moscow 117997, Russia; svkhokhlova@mail.ru (S.V.K.); g_sukhikh@oparina4.ru (G.T.S.); 3Institute of General Pathology and Pathophysiology, Moscow 125315, Russia; zolly_sten@mail.ru; 4“N.N. Blokhin National Medical Research Center of Oncology of Ministry of Health of the Russian Federation, Moscow 115522, Russia; mstenina@mail.ru

**Keywords:** breast cancer, circulating tumor DNA, liquid biopsy, digital PCR, next-generation sequencing, chemotherapy, prognosis, prediction, minimal residual disease, progression monitoring

## Abstract

This paper introduces the reader to the field of liquid biopsies and cell-free nucleic acids, focusing on circulating tumor DNA (ctDNA) in breast cancer (BC). BC is the most common type of cancer in women, and progress with regard to treatment has been made in recent years. Despite this, there remain a number of unresolved issues in the treatment of BC; in particular, early detection and diagnosis, reliable markers of response to treatment and for the prediction of recurrence and metastasis, especially for unfavorable subtypes, are needed. It is also important to identify biomarkers for the assessment of drug resistance and for disease monitoring. Our work is devoted to ctDNA, which may be such a marker. Here, we describe its main characteristics and potential applications in clinical oncology. This review considers the results of studies devoted to the analysis of the prognostic and predictive roles of various methods for the determination of ctDNA in BC patients. Currently known epigenetic changes in ctDNA with clinical significance are reviewed. The possibility of using ctDNA as a predictive and prognostic marker for monitoring BC and predicting the recurrence and metastasis of cancer is also discussed, which may become an important part of a precision approach to the treatment of BC.

## 1. Introduction

### 1.1. The Current State of the Problem of Breast Cancer around the World

Worldwide, breast cancer (BC) is the most frequently diagnosed cancer and the leading cause of cancer death among females. It accounts for 24.5% of cancer cases and 15.5% of cancer deaths in women. In 2020, there were 2.26 million new cases of BC, and 685,000 deaths from BC worldwide [1].

A number of factors are associated with an increased risk of developing BC (hereditary factors, hormonal factors, lifestyle factors, benign breast diseases, high radiographic density of the breast, environmental factors); however, most of these factors are associated with a moderate increase in individual risk, while at least half of the women who are diagnosed with BC have no identifiable risk factors, except for increasing age and belonging to the female sex. Several genes have been identified that predispose individuals to hereditary BC. The majority of them are *BRCA1* and *BRCA2* genes. The estimated lifetime risk of BC development in carriers of *BRCA1* and *BRCA2* mutations ranges from 26% to 85%. Other genetic mutations associated with BC risk have a lower penetrance. These are mutations in the *TP53*, *PTEN*, *CDH1*, *CHEK2*, *PALB2*, and *ATM* genes [2].

The prognosis and the treatment of BC are generally determined by the stage (i.e., the degree of anatomical spread of the tumor) and the biological characteristics of the tumor (i.e., whether it belongs to a specific molecular biological subtype).

In essence, BC is biologically a heterogeneous disease; individual subtypes are characterized by different sensitivities to drugs and different prognoses. The key characteristic that influences the choice of treatment tactics is the expression of hormonal receptors (estrogen (ER) and progesterone (PgR)) and human epidermal growth factor receptor 2 (HER2). These markers can be used to define different functional groups of tumors: hormone receptor-positive, HER2-negative tumors (sensitive to hormone therapy); HER2-overexpressing tumors with or without hormone receptor expression (responsive to anti-HER2 therapy); hormone receptor-negative, HER2-negative tumors (“triple-negative” tumors; not sensitive to hormone therapy and anti-HER2 therapy). Studies on the molecular biology of tumors have shown that these subtypes have different well-defined genomic profiles [3,4], which allows for subdividing BC into luminal A (ER-positive and PgR-positive/HER2-negative with lower-grade features), luminal B (ER-positive and/or PgR-positive but with higher-grade features or HER2-positive), HER2-positive (ER-negative/PR-negative/HER2-positive), and basal-like (ER-negative/PR-negative/HER2-negative). These subgroups affect both the likelihood and timing of cancer recurrence: triple-negative/basal-like, HER2-positive, and luminal B BC are at greater risk of early recurrence relative to luminal A cancers, which have a longer latency period of possible recurrence [5]. Thus, today, BC is not considered as one disease, and when establishing a diagnosis, in addition to determining the anatomical stage, it is necessary to determine the molecular biological subtype (using genetic testing or by determining ER, PgR, and HER2 in combination with the determination of the proliferation marker Ki67). In addition to defining biologic tumor subsets, gene expression profiling (MammaPrint, Irvine, USA, Oncotype DX, Redwood City, USA, Prosigna, South San Francisco, USA) has been used to stratify tumors as having good-risk or poor-risk prognostic signatures. Retrospective analyses suggest that these gene signatures contribute independent prognostic information above and beyond that achieved with the use of traditional pathologic markers, such as stage, grade, lymphovascular invasion, and ER/PgR/HER2 status. These genomic assays have proven especially valuable in distinguishing the prognosis within the subset of luminal, ER-positive, HER2-negative breast tumors, which are the most common form of BC [2]. However, these genomic assays cannot provide all the necessary information for a precision approach for all subtypes of BC.

### 1.2. Determination of Circulating Tumor DNA Is a New Approach

The discovery and implementation of prognostic and predictive tumor markers in everyday clinical practice, along with the possibility of individualization of therapy for BC, has given rise to a number of methodological problems. Firstly, BC is a heterogeneous disease, not only within the given nosological form (referencing the four molecular subtypes described previously), but also within a specific case of the disease, since one tumor node may contain foci with different expressions of ER, PgR, and HER2. Core biopsies, traditionally used for diagnostic purposes, do not guarantee the collection of representative material in the case of a heterogeneous tumor. Secondly, in the course of treatment and progression, a tumor can change its biological characteristics. For example, the sensitivity of a tumor to therapy changes due to the loss of receptors in metastatic and recurrent foci, as well as the acquisition of mutations (for example, *ESR1* during hormone therapy with aromatase inhibitors). This should be taken into account when planning treatment. Reobtaining material using a core biopsy, including from a metastatic focus in order to determine the biological characteristics of the tumor, is often difficult and sometimes technically impossible. In this regard, the method of liquid biopsy with the determination of various components, such as circulating tumor DNA (ctDNA) isolated directly from plasma or from extracellular vesicles, as well as circulating tumor cells, and others, is of great value, since it allows the use of a minimally invasive procedure (for example, blood sampling from a peripheral vein) to obtain diagnostic material as often as desired.

ctDNA detection is the most widely studied, minimally invasive alternative for the molecular characterization of solid tumors. It has now been shown that ctDNA is characterized by the presence of genetic and epigenetic changes identical to those contained in the tumor tissue. The determination of specific changes in driver genes may be of diagnostic value, correlating with the response to therapy and the dynamics of the tumor process during the treatment period. Since most of the molecular abnormalities found in plasma ctDNA reflect genetic and epigenetic changes in the primary tumor, ctDNA analysis is a convenient predictive and prognostic method for monitoring the course of oncological diseases. The key advantages of ctDNA analysis include its high specificity; correlations with tumor burden, including metastasis, and response to treatment; and the representativeness of the obtained material, which excludes to a maximum extent the possible heterogeneity of the tumor.

In this review, we provide a consistent introduction to the field of liquid biopsies, focusing on studies related to the determination of ctDNA in BC for various applications.

## 2. Circulating Tumor DNA: What Is It?

### 2.1. Circulating Tumor DNA Is a Variant of Liquid Biopsy

The use of a precision approach in oncology has increased significantly over the past few decades due to increased knowledge of cancer genomics and the identification of genetic tumor biomarkers [6]. The basis of research in this area is the study of tumor tissue. However, the use of traditional tumor biopsies (core biopsies) in clinical practice is associated with a number of limitations. First, tumor biopsies are invasive procedures that involve potential complications for patients and have limitations in taking tissue samples, especially from distant metastases localized in hard-to-reach parts of the body. Second, a traditional biopsy sample may not reflect the true characteristics of the tumor due to its heterogeneity and changes during the course of treatment and progression [7].

Liquid biopsies have emerged as a minimally invasive approach that allow the identification of important tumor-associated biomarkers throughout the course of the disease, including the detection of drug resistance/sensitivity mutations [8,9].

ctDNA is a widely studied variant of liquid biopsies. In addition to ctDNA, the concept of liquid biopsies includes the analysis of circulating tumor RNAs, long non-coding RNAs, mRNAs, and microRNAs, either directly or through isolation from extracellular vesicles (microvesicles, exosomes), as well as circulating tumor cells, or CTC (amount, content of proteins, DNA, mRNA and miRNA), tumor-educated platelets (alterations in their RNA profile), and proteins [8,10,11]. These markers are transferred from primary tumors and metastases to peripheral blood and other biological fluids. Liquid biopsy generally involves a blood test, but may also include the study of other biological fluids, such as urine, saliva, cerebrospinal fluid, and pleural, pericardial, and ascitic effusions [12,13]. In all of these listed body fluids, it is possible to determine the ctDNA biomarkers necessary for personalized therapy.

Liquid biopsies combined with highly sensitive molecular technologies can solve the problem of heterogeneity of a tumor, i.e., molecular abnormalities in metastases. This is often impossible with traditional biopsies, since such a sample may not contain the entire variety of tumor cells [14,15]. The minimal invasiveness of liquid biopsies allows them to be performed as often as required in a specific clinical situation, and therefore the method can be used for early diagnosis and screening [15,16,17], prognosis, early detection of disease recurrence (subclinical), detection of minimal residual disease (MRD), and monitoring of disease progression and response to treatment during neoadjuvant chemotherapy (NAC) and adjuvant therapy [18,19,20].

### 2.2. Circulating Tumor DNA and Its Main Features

ctDNA is part of cell-free nucleic acids (cfNAs). The first mention of cfNAs in the blood dates back to the middle of the last century. In 1948, Mandel and Métais isolated cfNAs from human plasma for the first time [21]. For a long time, research in this area has focused on autoimmune diseases, in which high levels of cfNAs have been found in the serum of patients. cfNAs, as a prognostic marker of cancer, were first mentioned in 1977. In a study by Leon et al., it was found that in the blood serum of patients with various types of cancer, the level of cfNAs was increased. Moreover, it was shown that the level of cfNAs decreased with a positive response of patients to radiation therapy. This suggests that serum cfNAs may be an important tool for evaluating the efficacy of therapy, including the comparison of different regimens [22]. In 1989, Stroun et al. found that part of the plasma DNA comes from cancer cells [23]. In 1999, M. Esteller et al. detected abnormal promoter methylation of tumor-associated genes in serum DNA in all stages of lung cancer development [24]. In the same year, Silva et al. found genetically altered and methylated cfNAs in the plasma of patients with BC [25,26,27]. As was shown later, ctDNA methylation can be informative in the early detection, prognosis, therapy response, and MRD detection in various types of cancer.

cfNAs currently include various variants of circulating extracellular nucleic acids: genomic DNA, mRNA, viral DNA and RNA, microRNA, other types of non-coding RNA, and mitochondrial DNA. Cell-free DNA (cfDNA) is a double- or single-stranded fragmented DNA, the length of which varies from 180 to 21 kb. cfDNA can be released by both normal and tumor cells into most body fluids including blood; urine; cerebrospinal fluid; ascitic fluid, etc. In the blood, cfDNA is present mainly in the form of nucleosomes, which are macromolecular complexes of histones and DNA [28] or vesicles (Figure 1) [29]. Such structures protect cfDNA from the action of nucleases and prevent an immune response to the presence of cfDNA in the blood [29]. cfDNA can also be bound to the surface of blood cells through specialized membrane proteins. In cancer patients, part of the cfDNA is represented by circulating tumor DNA (ctDNA), which is found in a variable but low percentage (0.01–1.0% or less) of the total amount of cfDNA (Figure 1b). This amount ranges from 5 to 1500 ng/mL and its variation depends on the stage of the tumor and its location [30]. cfDNA is present in healthy people, but its concentration is significantly lower compared to cancer patients due to active degradation by nucleases; the average amount for healthy people is about 5 ng/mL (1–10 ng/mL) [30,31].

The release of ctDNA into the blood occurs from a variety of sources, including the primary tumor, circulating tumor cells in the peripheral blood, and distant metastatic foci [32,33]. ctDNA can be released through a variety of mechanisms including apoptosis (fragments shorter than 200 bp enriched in tumor genomic changes), necrosis (fragments larger than 10,000 bp), ferroptosis, pyroptosis, oncosis, and phagocytosis, and also as a result of processes not associated with cell death, such as aging, active secretion into extracellular vesicles, and the excretion of mitochondrial DNA [34]. In the patient’s body, the half-life of ctDNA is short [35], which is convenient for a “real-time” approach for analyses of ctDNA for various therapeutic applications.

The basis of ctDNA usage for clinical application is the ability to quantify aberrant methylation and tumor-specific mutations in cfDNA. The number of tumor-specific mutations in cfDNA in cancer patients may vary. For example, more than 10% may be detected in a metastatic setting, while less than 0.1% may be detected in early-stage cancer or with minimal residual disease [36]. It is known that the amount of cfDNA in blood serum is 3–24 times higher than in plasma [37,38]. However, this higher amount is associated with the contamination of the DNA released by blood cells during coagulation, so plasma is recommended for ctDNA analysis [39]. Studies that have focused on the problem of processing procedures for plasma circulating cfDNA from cancer patients have shown that ctDNA levels are stable for 24 h at room temperature, or even 3 days when stored at +4 °C using EDTA tubes [40].

In this review, we focus on ctDNA as one of the markers in liquid biopsies. Since the advent of liquid biopsies, ctDNA has been used in the development of non-invasive, specific molecular assays to obtain diagnostic, therapeutic, and prognostic information [41,42,43]. In addition to studying ctDNA directly, the study of ctDNA together with CTC, and also, possibly, with their clusters in the form of circulating tumor microemboli (CTM) is currently promising [44,45,46,47,48]. In addition, the cell-free DNA Damage Index (cfDI) parameter can be used as a biomarker for identifying certain types of tumors, including BC, as well as monitoring the response to treatment [49,50,51].

### 2.3. Basic Methods for Studying ctDNA

There are several methodological approaches for the quantitative analysis of ctDNA. When choosing a method for studying ctDNA, the following parameters should be taken into account. The average amount of cfDNA in human plasma is usually about 5 ng/mL, varying from 1 to 10 ng/mL [30,31]. This corresponds to approximately 1500 conditional complete genomes per 1 mL of plasma. The ratio of tumor to normal DNA in plasma can be detected by searching tumor-specific mutations in cfDNA; this is called the mutant allele fraction (MAF). An MAF of 0.1% means that for every 999 cfDNA molecules from normal tissue, there is 1 ctDNA molecule. The MAF depends on the tumor size: the smaller the tumor, the lower the MAF. For a tumor size of approximately 10–12 mm, the MAF drops to 0.01% [52]. Thus, methods with high sensitivity and specificity should be used for the analysis of ctDNA.

The methods for studying ctDNA can be roughly divided firstly into targeted methods (PCR-based and targeted sequencing methods), in which the purpose is to target specific, previously discovered mutations, both single mutations and groups of mutations, and secondly, search methods (whole-genome sequencing (WGS), whole-exome sequencing, large panels of genes), which make it possible to evaluate a wide range of mutations up to the whole-genome/exome level (see Table 1; adapted from [53]). Epigenetic changes in ctDNA can be studied using both of these methodological approaches. However, to study epigenetic markers’ preliminary bisulfite conversion is necessary. In addition to these two main approaches, pyrosequencing and array-based genome-wide DNA methylation analysis can also be used to analyze methylation in ctDNA.

#### 2.3.1. PCR-Based Methods

PCR-based methods are used as targeted methods for ctDNA analysis. The real-time PCR method is very inexpensive and fast, but it has an extremely low sensitivity for detection of tumor-specific mutations in ctDNA, which makes it of little use in practice. Digital PCR (dPCR) is much more sensitive. Another advantage of dPCR is the direct determination of the concentration of the desired mutant variant without the need to construct a calibration curve. In dPCR, a DNA sample is distributed in microwells or droplets (digital droplet PCR, or ddPCR), resulting in thousands of parallel PCR reactions. dPCR can quantify the proportion of mutant variants against a background of wild-type cfDNA based on the amplification of one DNA molecule in a microwell/droplet, and it has a sensitivity of 0.1%. This makes it possible to use the dPCR method for ctDNA detection. A significant amount of work on the study of the clinical application of ctDNA has been carried out using the ddPCR-based approach [20,54,55,56,57]. Methyl-specific real-time PCR is quite sensitive to the methylated allele; therefore, in contrast to the analysis of mutations in ctDNA, determination of methylation is possible using a method based on real-time PCR. However, analysis using the methylation-specific digital droplet PCR (MS-ddPCR) method makes it possible to determine the presence of methylation in ctDNA with much greater sensitivity.

BEAMing (beads, emulsion, amplification, magnetics) is another PCR-based method used to detect known mutations. This approach combines the ddPCR method, in which the PCR reaction is carried out in a water–oil emulsion on magnetic beads, and flow cytometry as the detection method. Before the main reaction, the target sequences are enriched via standard PCR for amplification of regions of interest. Then, ddPCR is performed using primers covalently bound to magnetic beads. The target sequences amplified on magnetic particles are labeled with fluorescent probes specific for the mutant and non-mutant sequences and analyzed in a flow cytometer. As a result, the mutant is separated from the wild-type DNA, and the ratio of mutant DNA to wild-type DNA is determined [58]. Currently, a number of researchers are using this technology [59], which provides high sensitivity in the analysis of known mutations. However, this technique is less common than standard dPCR in microwells or droplets, due to the technique’s complex workflow. To study methylation in ctDNA, the methyl–BEAMing variation is used.

#### 2.3.2. NGS-Based Methods

Since the above methods allow only a limited number of mutations and epigenetic changes to be analyzed in a single experiment, the information about tumor heterogeneity presented in ctDNA may be lost. In addition, preliminary information about individual mutations in the tumor is required. Massive parallel sequencing or next-generation sequencing (NGS) techniques have been developed to address these issues. These methods are based on the genome-wide analysis of point mutations, copy number aberrations (CNAs), aberrant methylation, and other aberrations using WGS or whole-exome sequencing. This approach can be used to monitor mutations and changes in methylation during treatment, to detect de novo genetic and epigenetic changes including underlying therapy resistance, to identify new potent targets, and to characterize mutational burdens for the early detection of recurrence. The most informative approach for the search for tumor-specific mutations in cfDNA is the search for somatic mutations by comparing the results of the NGS of tumor tissue and blood. In this case, it is necessary to take into account the very important fact that most of the cfDNA originates from blood cells, and it is known that some of the somatic variants identified via NGS are the result of clonal hematopoiesis [60,61]. It is recommended that a blood cell fraction be included in NGS-based ctDNA testing to exclude mutations that occurred during clonal hematopoiesis and to avoid false-positive results [62].

Improvements in NGS technology have made it possible to screen wider genomic regions and simultaneously monitor multiple tumor-specific changes in a single analysis [63,64].

For the analysis of ctDNA based on NGS technology, various methodological approaches have been proposed, for example, standard amplicon sequencing (AmpliSeq), which uses oligonucleotide probes designed to target and capture regions of interest, followed by NGS, and various modifications of the NGS method. Personalized cancer profiling via deep sequencing or the CAPP-Seq method has been developed, based on the use of a set of frequently mutated sites, known driver mutations and the NGS technique, which has been used by a number of authors [65,66,67]. Additionally, besides point mutations, CAPP-Seq can detect other genetic changes, such as copy number variations, insertions, and deletions. Another NGS-based method for ctDNA analysis is Safe-SeqS. It was designed to reduce the error rate during NGS and increase the sensitivity to rare mutations. This method is based on the fact that each DNA template molecule is encoded by a unique molecular identifier [68]. A method called targeted digital sequencing (TARDIS) was developed specifically to increase the sensitivity of residual disease detection during or after the completion of treatment in non-metastatic cancer patients. The authors used the simultaneous deep sequencing of patient-specific somatic mutations [69]. In addition, a search has been conducted for methodological approaches to NGS and the development of a custom pipeline to increase the sensitivity of the analysis [18].

The detection of specific, previously discovered mutations is possible using methods of targeted sequencing of an individual genes-based NGS technique. This approach is also being used to explore the clinical applications of ctDNA [54,70].

Aberrant methylation studies can be conducted via bisulfite sequencing, and multiplexed targeted NGS. All of them, like the PCR-based methods, require bisulfite conversion of the sample [71].

#### 2.3.3. Combined and New Approaches

New, combined approaches are being developed to increase the sensitivity of ctDNA detection. In particular, attempts have been made to increase the sensitivity of the CAPP-Seq method by enriching target regions via PCR [72], including the development of an approach for integrated digital error suppression [73].

Combining the NGS method with microfluidic technologies has led to the development of the TAM-Seq method, which allows whole-gene sequencing to detect tumor-specific mutations in cfDNA [74,75].

The development of microfluidic technologies makes it possible to use this approach to create platforms for ctDNA analysis [76]. A promising direction of new methodological approaches is based on electrochemical biosensors, which have recently been actively developed [77].

In the future, an improved version of nanopore technology can be used to study DNA methylation profiles in cancer patients. To date, this technology platform has some limitations in detecting specific mutations, but allows the sequence of methylated regulatory marks without special sample preparation despite limitations such as multiple nucleotide signals [78]. It may have several advantages for the study of DNA methylation profiles in future applications, since it can eliminate the need for bisulfite conversion. This technology provides the opportunity to rapidly study non-CpG DNA methylation, an area that has recently begun to develop. Thus, the development of modern technologies will make it possible, in the near future, to expect the creation of a cost-effective approach for the analysis of ctDNA. Efforts are currently being made to develop commercial assays for ctDNA analysis. In particular, for BC, there is the Oncomine breast cancer cfDNA test (Thermofisher, Waltham, MA, USA) based on AmpliSeq technology, which includes several genes.

### 2.4. Possible Applications of ctDNA in Oncology

Despite the extensive scientific research devoted to the study of cfDNA in various types of cancer [22,23,24,25,26,27], for a long time, there were no attempts to use it in clinical practice. In 2008, Diehl F. et al. studied ctDNA in 18 colorectal cancer patients and found some mutations of genes such as *APC*, *KRAS*, *TP53*, and *PIK3CA*, while the frequency of ctDNA mutations changed during treatment [35]. Thus, using the example of patients with colorectal cancer, the potential clinical significance of ctDNA determination was shown for the first time. ctDNA can be found in many types of cancer, including BC [79], lung cancer [80], colorectal cancer [81], prostate cancer [82], gastric cancer [83], ovarian cancer [84], and others [85,86,87,88,89]. It has also been shown that ctDNA can be detected in early-stage cancer patients, while the most characteristic change in cancer patients is DNA hypermethylation. A significant number of works have revealed hypermethylation of ctDNA in lung cancer and colorectal cancer, which makes it possible to distinguish patients from healthy people [90,91,92,93,94,95]. In addition, an increase in methylation has been shown in both endometrial and gastric cancer [77,96]. This allows for ctDNA to be considered as a biomarker for the development of non-invasive methods for the early detection of cancer, for which new methodological approaches are currently being developed [76,77,97,98].

It has also been shown that ctDNA has predictive properties. Most of these studies have been carried out on lung cancer [80,99,100,101,102] and colorectal cancer [81,103], including the use of DNA methylation analysis [104]. However, other types of cancer, such as ovarian cancer [84], prostate cancer [82,105], pancreatic ductal adenocarcinoma [106], melanoma [107], and others [88,89,108,109,110], have also been widely represented in the works carried out in recent years.

The analysis of ctDNA allows for monitoring the response to treatment in real-time regimens, including the detection of molecular residual disease, which has been described for lung cancer [101,111,112,113], colorectal cancer [114], ovarian cancer [115], and others [87,116,117]. It was found that the use of the methylation markers for a number of genes allows not only for the identification of patients with colorectal cancer, but also for monitoring recurrence [95]. Thus, epigenetic markers in ctDNA, and in particular, methylation changes in specific genes or regions that are reflected in ctDNA, can be used for the early detection and diagnosis of a number of cancers, and for treatment monitoring.

Changes in ctDNA levels correlate with tumor volume, allowing their use as a method for the minimally invasive monitoring of tumors’ response to treatment in many types of cancer [80,118,119]. A successful attempt was made to use ctDNA to detect and monitor primary and metastatic pediatric brain cancer [120]. In the case of renal cancer, lower plasma ctDNA levels relative to other cancers of a similar size and stage were shown; nevertheless, the authors concluded that ctDNA still has potential clinical utility in the management of patients with renal tumors [121].

Thus, the study of ctDNA has many potential applications in oncology, such as early diagnosis, tumor molecular profiling, early detection of resistance mutations, assessment of response to treatment, assessment of minimal residual disease, and progression monitoring.

## 3. Clinical Value of ctDNA in Breast Cancer Patients

The actual clinical tasks in the treatment of BC are early diagnosis, monitoring of disease progression, the evaluation of effectiveness, including in the case of NAC, the detection of minimal residual disease, and the prediction of the effectiveness of different treatment methods.

### 3.1. Application of ctDNA for Early Detection/Screening of Breast Cancer

Currently, the only available methods for the screening and early detection of localized BC, for which there is a possibility of radical treatment, are self-examination and objective imaging methods such as mammography, ultrasound, and magnetic resonance imaging. However, mammography has age restrictions, as it has been proved to be generally not informative in young women. In addition, mammographic screening is associated with overdiagnosis. All this requires the development of new diagnostic tests with fundamentally different capabilities, and as a result, liquid biopsies providing the determination of ctDNA may well claim the role of a screening method for BC. There are a number of studies that substantiate the possibility of using ctDNA analysis for the early detection of BC. Some works are based on the identification of mutations in ctDNA [122,123], while other works are based on the analysis of aberrant methylation in patients compared to healthy people [17,124]. Kamel A.M. et al. used parameters of DNA damage and determined that the level of DNA damage reliably identifies patients with BC compared to benign breast patients and healthy subjects (*p* < 0.001) [49]. The determination of mutations in genes using the CancerSEEK panel made it possible to propose a method that detects BC with a sensitivity of 33% and a specificity of 99% [122]. A use case for ctDNA was proposed by Rodriguez B.J. et al. in 2019, which consisted of the detection of mutations in the *TP53* and *PIK3CA* genes [125]. The development of a panel-based approach including exonic regions of 33 genes involved in BC pathogenesis yielded a specificity of 86.36% and a positive predictive value of 88.46% for BC detection [123].

Li Z. et al. were the first to evaluate the methylation status of the *EGFR* and *PPM1E* promoters in the detection of BC. They observed that patients with BC had significantly higher levels of methylation than healthy individuals [126]. Fackler M.J. et al. and Visvanathan K. et al. have consistently developed assays to distinguish BC from benign breast disease and healthy normal subjects using ctDNA (the earlier cMethDNA and the more recent automated Liquid Biopsy for Breast Cancer Methylation (LBx-BCM) prototype based on the previous one). LBx-BCM achieved a ROC AUC = 0.909 (95% CI = 0.836–0.982), 83% sensitivity and 92% specificity; cMethDNA achieved a ROC AUC = 0.896 (95% CI = 0.817–0.974), 83% sensitivity and 92% specificity [42,127,128,129]. Approaches based on whole-genome bisulfite sequencing analysis have been introduced in recent years, making it possible to obtain significant results. In one of these works, Gao Y. et al. exhibited high accuracy in early (AUC of 0.967) and advanced (AUC of 0.971) BC stages [17]. Hai L. et al. identified potential breast-cancer-specific methylation CpG site biomarkers with high specificity and sensitivity [130]. In a recent work, the genome-wide methylation method was enhanced with additional analyses of copy number alterations and four-nucleotide oligomer end motifs (4-mers), as well as multi-featured machine learning. As a result, the authors obtained a model that achieved an AUC of 0.91 (95% CI 0.87–0.95) and a sensitivity of 65% at 96% specificity [124]. Methylation changes in breast cancer-associated genes or regions can be detected in the blood in the earliest stages of the disease. This opens up the potential for non-invasive screening approaches that could complement or improve existing diagnostic methods.

Thus, ctDNA can be considered as a potential biomarker in liquid biopsy samples to detect aberrant methylation and specific mutations in BC, in order to develop non-invasive methods for the early detection of BC.

### 3.2. ctDNA as a Predictive and Prognostic Marker in Breast Cancer Treatment

This review is focused on research devoted to the field of liquid biopsies and analysis of ctDNA in patients with BC for different therapeutic applications. This part of the review is based on clinical trials conducted over the past few years, as well as important earlier works on the use of ctDNA for the treatment of BC.

#### 3.2.1. Two Main Approaches to Determining ctDNA in Breast Cancer

Analysis of the results of recent clinical studies in BC allowed us to identify two main methodological approaches in determining ctDNA for use in the treatment of BC (Table 2). The first is the analysis of point mutations and aberrant methylation in the presence of known driver genes, which is more relevant for hormone-dependent and HER2-positive BC. The second approach is based on whole-genome or -exome analysis of cfDNA. This variant is more convenient for triple-negative BC because there are no driver genes for analysis in this subtype of BC.

#### 3.2.2. Clinical Value of ctDNA Determination for Breast Cancer Treatment

From the point of view of clinical value, the considered studies can be divided into three main groups: studies devoted to the early detection of recurrence or progression of BC, the detection of MRD after primary treatment of BC, and usage in metastatic BC as a prognostic criterion. All of these lines of research are promising for the future development of protocols and assays for clinical use.

##### Early Detection of Recurrence/Progression of Breast Cancer

The significance has been shown of ctDNA in BC in the detection of preclinical metastases and the prediction of recurrence after surgery and/or NAC in patients with specific hotspot mutations [150,155], chromosomal rearrangements [156], and amplifications [157].

The most common driver genes in the tissues of primary and metastatic breast tumors—in particular, the *TP53* gene of the p53 tumor protein, and the *PIK3CA* gene and others—can be analyzed using liquid biopsies. Using the panel, which includes 52 of the most common oncogenes and tumor suppressors, the detection rate of tumor-specific mutations in ctDNA was 37% and 81% for I–III stage BC and metastatic/recurrent BC, respectively. Of the 57 single nucleotide variants (SNVs) detected, the majority of the variants were found in five genes, namely *TP53* (37%), *PIK3CA* (21%), *AKT1* (7%), *EGFR* (5%), and *KRAS* (5%). The frequency of mutations’ detection in ctDNA correlated with the stage of the disease, increasing with metastatic or recurrent disease (*p* = 0.00026), with lymph node involvement (*p* = 0.00649), and with distant metastases (*p* = 0.0005) [61]. Significantly lower levels of promoter promA methylation of *ESR1* gene at baseline were observed in patients with liver metastases (*p* = 0.0212) in ER-positive metastatic BC [131]. 

Riva F. et al. found *TP53* mutations in patients’ ctDNA at baseline in 75% triple-negative BC patients. Its detection was associated with the mitotic index (*p* = 0.003), tumor grade (*p* = 0.003), and stage (*p* = 0.03). The patient with rising ctDNA levels experienced tumor progression during NAC [54].

As can be seen from the frequency of detection of driver mutations in ctDNA, a significant number of BC patients do not have hotspot mutations and therefore cannot be controlled using the driver gene approach. Therefore, a comprehensive solution is needed based on the use of whole-genome/exome sequencing of each patient’s tumor to search for characteristic somatic mutations. An intermediate approach was used in the c-TRAK TN trial, enrolling patients with triple-negative BC. Since these patients did not have known target genes, the analyses of tumor samples were carried out via targeted sequencing, principally with gene sequencing panels. However, for analysis in cfDNA, only one or two mutations from the sequencing were selected. ctDNA-positive patients accounted for 27.3% of all the patients over the 12 months of observation. The patients had a high rate of metastatic disease in ctDNA-positive detection (nearly 72%) [132].

Using ultra-deep sequencing to detect tumor-specific mutations in cfDNA selected from data from the whole-exome sequencing of primary tumors, the sensitivity and specificity of this parameter in relation to the incidence of recurrence in patients with BC has been revealed [151]. Tumor-specific mutations in ctDNA were found to be associated with the development of disease recurrence and progression-free survival (PFS) or disease-free survival (DFS) [18,19,133,150]. Parsons H.A. et al. developed an ultra-sensitive, patient-specific ctDNA analysis based on exome sequencing to identify patient-specific SNVs, which can be used to design custom ctDNA tests. ctDNA detection was strongly associated with distant recurrence (HR = 20.8; 95% CI 7.3–58.9). The median lead time from the first positive sample to recurrence was 18.9 months (range = 3.4–39.2 months). It is important to note that most patients had a much lower number of mutations in the tumor for analysis in ctDNA than the test allowed (median 57, range 2–346) [18].

Aberrant methylation in the specific region EFC#93 (a pattern of five linked CpGs methylated in BC) was an independent and poor prognostic marker in pre-chemotherapy samples for death (HR = 7.689), DFS, and OS. EFC#93 positivity after chemotherapy was significantly more frequent at late stages (T2–T4) (*p* = 0.014) [46].

From a clinical point of view, data on the relationship between ctDNA and tumor burden are extremely important (Table 3). The data allow us to discuss the prospect of using this test for the early detection of recurrence. In some patients, even in the early stages of BC, a relapse of the disease develops after primary treatment. The risk of relapse weighs heavily on the minds of both patients and physicians for years after treatment of the primary tumor is complete. Regular follow-up with a medical oncologist aims to identify new symptoms or changes on physical examination. The current National Comprehensive Cancer Network (NCCN) and American Society of Clinical Oncology (ASCO) guidelines recommend the taking of a routine history and physical examination with a certain frequency. Both the NCCN and ASCO do not recommend the use of most of the examinations and blood tests (e.g., routine complete blood counts, chemistry panels, tumor markers, bone scans, computed tomography (CT) scans, magnetic resonance imaging (MRI) scans, positron emission computed tomography (PET) scans, or ultrasound examinations) in asymptomatic patients without specific clinical examination findings. Dedicated breast MRI may be considered for post-therapy surveillance in women at high risk for bilateral disease, such as carriers of the *BRCA 1/2* mutations [158,159]. However, despite these guidelines many patients receive high-cost imaging analysis (CT, brain or body MRI, PET, and bone scans) and tumor marker blood tests during routine follow-up exams, exposing them to radiation and increasing healthcare costs. Although a number of past studies of more thorough and more financially costly dynamic follow-up (compared to the standard one) has not revealed advantages in the PFS and overall survival (OS) of patients with early detection of metastatic disease, the early detection of locoregional relapses allows for radical treatment and has a positive effect on long-term results [160,161]. As for metastatic disease, technologies for the local control of metastatic foci (stereotactic radiation therapy, radiofrequency, cryoablation, etc.) are currently available, which, in combination with highly effective methods of drug therapy, can significantly improve long-term treatment results. However, the scope of local methods is limited to clinical situations with a few (single) foci of small sizes (oligometastatic disease), which increases the demand for early detection of metastatic processes. Over the past decade, the development of non-invasive biomarker assays has enabled the low-cost early detection of cancer: according to the available data, the appearance of ctDNA may precede the clinical manifestation of metastatic disease by many months. Attempts are being made to develop automated assays for monitoring therapeutic response and predicting disease recurrence based on the analysis of aberrant methylation of a panel of genes [129].

##### ctDNA as a Surrogate Marker for Minimal Residual Disease after Primary Treatment for Breast Cancer

Neoadjuvant drug therapy is an essential treatment for early and locally advanced BC. As is known, in early and locally advanced BC, especially in triple-negative and HER2-positive variants, the achievement of a pathological complete response (pCR) has important prognostic value and is associated with an increase in OS and PFS [162,163,164,165,166,167,168,169].

In clinical practice, to assess the effectiveness of NAC, traditional clinical and instrumental methods (examination, palpation, ultrasound of the mammary glands and regional zones, mammography) are currently used, which often do not allow for the true effect of treatment to be evaluated. For example, in the study of Ignatova E.O. et al., when analyzing the results of NAC in 41 patients with early primary operable triple-negative BC, it turned out that in the clinical and instrumental assessment of the effect of using ultrasound and mammography, the frequency of complete responses was only 30%, while the pathomorphological study of the postoperative material revealed a pCR in 65% of cases [170]. Generally, six to eight courses of chemotherapy ± targeted therapy (anti-HER2 therapy if necessary) are used in NAC. However, for some patients in the early stages with a high sensitivity to drug therapy, such a volume of treatment may be excessive, and analysis of ctDNA during treatment can help to find the optimal number of courses that each individual patient needs.

To assess the degree of pathologic response in everyday clinical practice, routine pathomorphological examination with the definition of categories, such as ypTypN (evaluation of tumor T and N categories after NAC through pathological staging) and RCB (Residual Cancer Burden), is currently used. However, this method does not allow for studying 100% of the removed tissues and therefore carries the possibility of error due to the heterogeneity of the diagnostic material. Meanwhile, today, information on the degree of pathologic response of the tumor is the key to determining the tactics of post-neoadjuvant therapy in triple-negative BC and HER2-positive BC.

In addition, an extremely interesting modern trend in the treatment of early BC is the study of the possibility of refusing surgical treatment in case of pCR achievement. In such studies, a biopsy of tumor tissue is used to confirm a pCR; however, this method does not allow for analyzing the entire mass of the tumor.

Recent clinical studies have shown that ctDNA may play a role in detecting MRD and early detection of resistance to therapy, i.e., the molecular recurrence of BC [132,150,152,155]. It can also be used as a method for monitoring disease progression in patients with advanced BC [19,171,172]. A relationship was found between *TP53* mutations in triple-negative BC patients’ ctDNA and the effectiveness of NAC. ctDNA positivity after one cycle of NAC correlated with shorter DFS (*p* < 0.001) and OS (*p* = 0.006) [54]. Based on the whole-exome detection of mutations in tumor tissue and the formation of individual mutation panels for each patient, one of the recent studies revealed an association between the presence of tumor-specific mutations in ctDNA and response to NAC, the risk of recurrence, metastasis, and 3-year survival. It was revealed that an important time point for the prognosis of pCR is in the early (3 weeks from the start) period of NAC. At T0 (pretreatment), 61 of 84 (73%) patients were ctDNA-positive, which decreased over time (T1: 35%; T2: 14%; and T3: 9%). Patients who remained ctDNA-positive at T1 (3 weeks after the initiation of paclitaxel) were significantly more likely to have residual disease after NAC (83% non-pCR) compared with those who cleared ctDNA (52% non-pCR; OR = 4.33, *p* = 0.012). After NAC, all patients who achieved a pCR were ctDNA-negative (n = 17, 100%). For those who did not achieve a pCR (n = 43), ctDNA-positive patients (14%) had a significantly increased risk of metastatic recurrence (HR = 10.4). Interestingly, patients who did not achieve a pCR but were ctDNA-negative (86%) had an excellent outcome, similar to those who achieved a pCR (HR = 1.4). In this trial, the lack of ctDNA clearance (absence of tumor-specific mutations in ctDNA) was a significant predictor of poor response and metastatic recurrence (HR = 22.4; 95% CI 2.5–201, *p* < 0.001), while clearance was associated with improved survival even in patients who did not achieve a pCR [19]. Therefore, the personalized monitoring of ctDNA during NAC may aid in the real-time assessment of treatment response and help fine-tune pCR as a surrogate endpoint of survival [19,69].

ctDNA methylation analysis may also be a marker of treatment response [134,135,136]. A significant correlation was found with the extent of residual tumor burden, and response to therapy with the methylation of the *RASSF1A* promoter region in ctDNA after NAC [134,136], but not for *MGMT* promoter [137]. A systematic review on the prognostic potential of cfDNA methylation for hormone receptor-positive BC was conducted in 2020 and used most of the previous studies. Hypermethylation of the markers *RASSF1*, *BRCA*, *PITX2*, *CDH1*, *RARB*, *PCDH10* and *PGR*, and *GSTP1* showed a statistically significant correlation with poor disease outcome [173]. The authors also identified high heterogeneity in the protocols of these studies, which indicates the need for standardized assessment methods.

The detection of MRD can be critical in assessing patients’ therapeutic response and subsequent treatment decisions. Several studies have evaluated the value of ctDNA in detecting MRD after NAC and surgical treatment [18,79]. Detection of ctDNA in serial samples using early-stage BC patients who received NAC before surgery was predictive of early relapse (HR = 12.0). Of the patients who did not relapse, 96% did not have ctDNA detected *p* < 0.0038) [150]. These results were confirmed by later studies of different subgroups of BC patients [138,153]. In the c-TRAK TN trial, the patients had a high rate of metastatic disease in ctDNA-positive detection (nearly 72%). An important conclusion was also drawn that three-month blood sampling is not frequent enough; this indicates the need for commencing ctDNA testing early, with frequent ctDNA testing regimes [132]. The presence of methylated ctDNA after the end of the chemotherapy was an indication of MRD [43].

ctDNA persistence during NAC and after primary treatment is associated with an increased risk of relapse and poor prognosis (Table 4). The arsenal of post-NAC methods has been replenished in recent years with new highly effective drugs (TD-M1 for HER2-positive BC; CDK4/6 inhibitors for hormone-dependent, HER2-negative BC; capecitabine and immunotherapy for triple-negative BC; olaparib for BRCA-associated BC). However, these highly effective drugs do not provide a cure for all patients and have significant toxicity, along with a high financial burden, requiring more precise individualization of treatment. The detection of ctDNA after NAC and surgical treatment could help identify a subgroup of patients who are indicated for one or another type post-NAC (taking into account the biological characteristics of ctDNA).

Finally, mutations in ctDNA can also be very useful biomarkers in the adjuvant therapy of BC, which is carried out in the absence of a clinically detectable tumor, i.e., virtually blind, as well as in the treatment of metastatic disease.

##### ctDNA Mutations as Prognostic and Predictive Factors for the Effectiveness of Therapy

Several studies have shown the usefulness of analyzing point mutations in the ctDNA of HER2-positive patients [139,140,141]. *TP53* mutations in ctDNA can be used as biomarkers of anti-HER2 antibody drug resistance in HER2-positive patients and HER2 tyrosine kinase inhibitor resistance in patients with *HER2* mutations without amplification. Patients carrying a *TP53* mutation in their ctDNA had a shorter PFS in response to anti-HER2 antibody treatment than those without a *TP53* mutation (HR = 1.42, *p* = 0.004). *TP53* mutations predicted the low efficacy of HER2 tyrosine kinase inhibitors (HR 2.83, *p* = 0.01) [139]. This is consistent with the data obtained by Rothé F. et al. [141].

Analysis of *HER2* amplification using whole-genome sequencing data for gene copy number variation analysis showed a high concordance rate of the observed status of *HER2* between tissue and ctDNA and a relationship with PFS and objective response during anticancer therapy. Among the patients that were histopathologically HER2-positive in the primary tumor, high-level amplification in *HER2* copy numbers of ctDNA in the baseline treatment was significantly correlated with the best objective response during anticancer therapy (*p* = 0.010). Moreover, *HER2* copy numbers fluctuated with the HER2-targeted therapeutic response, and the patients with a constantly positive level after 6 weeks of treatment appeared to suffer from a significantly reduced PFS (*p* < 0.001) [140].

In contrast to early, non-metastatic BC, ctDNA is detectable in the majority of metastatic BC: 85.71% of stage IV/M1 patients carried tumor-derived mutations in their blood, compared to only 57.81% of stage I–III/M0 patients [174]. A lot of studies dedicated to using ctDNA analysis in recurrent or metastatic BC have been conducted (Table 5). Most of them have used a point-mutation-based approach for ctDNA analysis. Many of these studies were aimed at exploring the possibility of using the ctDNA marker to evaluate the effectiveness of different types of therapy in this group of patients. It is known that patients with advanced BC have undergone prior aromatase inhibitor therapy followed by a selection of *ESR1* mutations by the time of tumor progression; therefore, attempts are being made to find new effective combinations for their treatment. Mutations of the main driver genes *TP53*, *ESR1*, *PI3KCA*, *HER2*, and *AKT1* are of the greatest significance, according to the available studies.

It has been observed that *TP53* mutations in exons 5–8 may be an independent prognostic marker for short DFS (HR = 1.50, *p* = 0.009), and *TP53* mutations may influence the effect of trastuzumab-based (anti-HER2) chemotherapy alone or in combination with taxanes [70]. Patients with *TP53* mutations had significantly worse OS than the carriers of wild-type alleles (*p* = 0.0094) [142]. In the case of treatment with palbociclib (CDK4/6 inhibitor) plus fulvestrant (antiestrogen), *PIK3CA* or *TP53* mutations were prognostic for OS (HR = 1.44 and 2.19, respectively) and associated with shorter OS (OR = 0.55 and 0.23) [143]. 

The presence of *ESR1* mutations in ctDNA of advanced BC patients showed a worse PFS compared with those with the *ESR1* wild type (HR = 1.46, *p* = 0.02) [20], and were highly associated with shorter OS (OR = 0.36) [143]. Dynamic monitoring of *ESR1* mutations served as a predictive biomarker of acquired resistance to endocrine therapy, whereas the combination of everolimus (inhibitor mTOR) with acquired *ESR1* mutations showed a longer PFS than other therapies without everolimus [142]. Patients with an *ESR1* mutation in their ctDNA had a worse PFS than those with the *ESR1* wild type when treated with aromatase inhibitor exemestane (2.6 months versus 8.0 months, HR = 2.12; *p* = 0.01), but not fulvestrant (HR 0.52; 95% CI 0.30–0.92; *p* = 0.02) [20]. In the case of palbociclib (CDK4/6 inhibitor) plus fulvestrant, this combination provided a PFS benefit regardless of the *ESR1* mutation status in ctDNA at day 1 or at the end of treatment [143]. But an early change from an aromatase inhibitor plus palbociclib to fulvestrant plus palbociclib treatment for patients with *ESR1* mutations in ctDNA ensured a gain in PFS [55,144]. So, this suggests the clinical efficacy of periodic monitoring of emerging or rising *ESR1* mutations in ctDNA to trigger an early change from an aromatase inhibitor plus palbociclib to fulvestrant plus palbociclib treatment in patients with rising circulating *ESR1* mutations detected even without tumor progression.

Recently, a numerically greater improvement in median PFS for patients with *ESR1* or *PIK3CA* mutations in their ctDNA was observed in the combination of abemaciclib (CDK4/6 inhibitor) plus fulvestrant [56]. The *PIK3CA* mutation status based on ctDNA also demonstrated prognostic significance for combination therapy with buparlisib (PI3K inhibitor) versus fulvestrant alone. Median PFS was significantly longer in the buparlisib versus placebo group [59].

The combination of a dual AKT and p70 ribosomal protein S6 kinase (p70S6K) inhibitor (LY2780301) with paclitaxel (taxane) in advanced BC was studied in patients with driver mutations of *PI3KCA*, *AKT1*, and *TP53* in their tumors. The presence of ctDNA upon inclusion was correlated with PFS (6-month PFS was 92% for ctDNA-negative patients versus 68% for ctDNA-positive cases (HR = 3.45, *p* = 0.007)). Copy number alterations were associated with disease progression under paclitaxel-LY2780301. Therefore, ctDNA detection at baseline was associated with shorter PFS, while plasma-based copy number analysis helped to identify alterations involved in resistance to AKT/p70S6K inhibitor plus paclitaxel treatment [57].

The PlasmaMATCH study, which was a trial of ctDNA testing in advanced BC patients, was focused on a number of driver genes for BC. The study included 1034 patients with disease progression who were stratified based on mutations in the *PIK3CA*, *ESR1*, *HER2*, *PTEN*, and *AKT1* genes identified in ctDNA. As a result, it was found that ctDNA testing enables the selection of mutation-directed therapies for patients with BC, with sufficient clinical validity for adoption in routine clinical practice. A positive response was demonstrated in carriers of *HER2* mutations treated with neratinib and, if ER-positive, with fulvestrant, as well as in carriers of *AKT1* mutations and ER-positive BC treated with capivasertib plus fulvestrant. In the study, neratinib for HER2-mutant BC identified via ctDNA testing had comparable activity to that observed when guided by tissue testing in a previous study [175], with durable responses. Similarly, capivasertib had high activity in patients with ctDNA-identified *AKT1* mutations, both in hormone-receptor-positive BC with fulvestrant and in hormone-receptor-negative BC as a single agent, confirming the results of a previous phase I study [176]. Therefore, the results demonstrated the clinically relevant activity of targeted therapies against rare *HER2* and *AKT1* mutations identified via ctDNA testing [145]. The effectiveness of treatment in accordance with the mutational status of ctDNA was also shown by Lyu D. et al.; the risk of progression was lower in groups of patients who received precision therapy based on ctDNA analysis (HR = 0.55, *p* = 0.023) [146]. Thus, the approach using known driver genes is convenient for hormone-dependent advanced BC.

A study using plasma samples from the CAMELLIA trial also confirmed the value of ctDNA analysis as a potential biomarker of therapeutic response and prognosis in patients with metastatic BC. The work was conducted using target-capture deep sequencing of 1021 genes to detect somatic variants in ctDNA. These data were used to determine the molecular tumor burden index (mTBI), which is a function based on somatic variations in ctDNA, considering the heterogeneity and dynamic evolution of the tumor. Patients with a high-level pretreatment mTBI had shorter OS than patients with a low-level pretreatment mTBI (*p* = 0.011). Patients with an mTBI decrease to less than 0.02% at the first tumor evaluation had longer PFS and OS (*p* < 0.001 and *p* = 0.007, respectively). The patients classified as molecular responders had longer PFS and OS than the nonmolecular responders (*p* < 0.001 and *p* = 0.036, respectively) [154].

Epigenetic changes were also studied in metastatic BC as a marker for predicting survival and disease outcome. Methylation of a number of genes separately (*ESR1*, *SOX17*) or as part of gene panels was significantly correlated to shorter OS and/or no pharmacotherapy response [128,147,148,149].

Changes in ctDNA levels correlate with tumor volume, making ctDNA an excellent non-invasive tool for monitoring response to treatment and determining the prognosis. These circumstances allow for this method to be considered as a convenient way of obtaining diagnostic material not only at the stage of primary diagnosis, but also during the treatment process. ctDNA analysis can provide a wide range of information about the processes in tumors; in particular, it can provide a rapid diagnosis of disease recurrence in previously treated patients (see Section 3.2.1), and can identify existing and newly emerged gene mutations that are targets of anticancer therapy (*PTEN*, *PIK3CA*, *ESR1*, *AKT*, *HER2*), which can help in selecting the best therapy for each patient. Meta-analysis has confirmed the role of ctDNA as a prognostic marker for patients with BC [177]. ctDNA analysis also provides information on clonal evolution and tumor heterogeneity and can be a useful tool in studying the mechanisms of resistance to therapy in metastatic BC [172].

## 4. Conclusions

Thus, the results of studying ctDNA in various clinical situations do not raise doubts about the high scientific and practical value of this method, which makes it possible to obtain important prognostic and predictive information necessary for the further individualization of the treatment of early and metastatic breast cancer. ctDNA methylation has been studied more extensively for the early detection and diagnosis of breast cancer compared to a prognostic or therapy monitoring marker.

The undoubted value of the method is the simplicity of obtaining biomaterial (peripheral blood, other biological fluids) that does not require complex invasive interventions, which theoretically (with the availability of methods for determining ctDNA) allows for the conduct of studies with any necessary frequency.

The determination of ctDNA makes it possible to level out errors associated with tumor heterogeneity and that inevitably arise in the study of a limited volume of tumor tissue, taken, as a rule, from one locus.

The main limitation for the widespread use of ctDNA analysis at the present stage is, of course, the high cost of the method for determining ctDNA, especially in the case of whole-genome sequencing, which makes it difficult to use it in clinical trials and to accumulate one’s own experience. However, with the development of biomedical technologies, this obstacle will certainly be overcome. In addition, in order for the results of relevant studies to be compared and analyzed, it is required to unify the methodology for determining ctDNA, including units of measure.

Finally, for the wide introduction of the method for the determination of ctDNA, its validation is required. The highest number of studies that we analyzed in this review were retrospective analyses and, of course, provided us with extremely important and interesting information, however, the value of the technique in terms of its impact on the results of treatment of patients with breast cancer should be proven in appropriate randomized clinical trials.

## Figures and Tables

**Figure 1 ijms-24-17073-f001:**
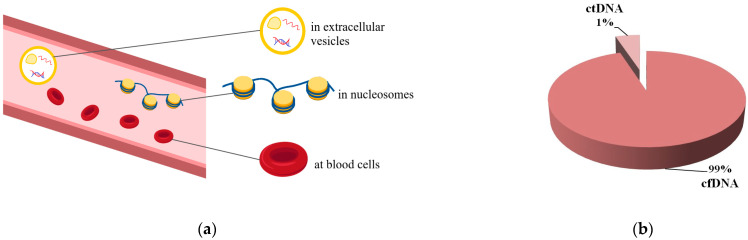
Cell-free DNA (cfDNA) in the blood: (**a**) formation in blood; (**b**) the amount of circulating tumor DNA (ctDNA) in cfDNA.

**Table 1 ijms-24-17073-t001:** Circulating tumor DNA analysis methods (adapted from [53]).

Target	Method	Example Assay	Sensitivity (%)	Advantages	Limitations
Single locus	Digital PCR	ddPCR, BEAMing	0.01	High sensitivity	Detects only known mutations
				Low DNA input	
				Provides quantification and monitoring of recurrent mutations	
Gene panel	Targeted panel sequencing	TAM-Seq, Safe-SeqS	0.01–1	High sensitivity	Less comprehensive than other NGS methods
				Fast	
				Cost-effective compared with other NGS methods	
	Targeted digital sequencing	TARDIS	0.03–1	High sensitivity	More complex workflow
	Hybrid capture sequencing	CAPP-Seq	0.02	Able to detect copy number variations and rearrangements	Requires high cfDNA input
					Less comprehensive
					More complex workflow
Comprehensive	Whole-exome sequencing Whole-genome sequencing		1–10	Identifies novel mutations	Less sensitive
				Does not require prior information about the tumor mutation	Expensive
					Longer turnaround time

**Table 2 ijms-24-17073-t002:** Two main approaches to determining ctDNA in clinical studies of breast cancer.

Authors	Date of Publication	Analysis	Reference
ctDNA Studies Using Point Mutation Analysis			
Fribbens, C. et al.	2016	*ESR1* mutations	[20]
Riva, F. et al.	2017	*TP53* mutations	[54]
Tolaney, S. et al.	2022	*PIK3CA* and *ESR1* mutations	[56]
Sabatier, R. et al.	2022	*PI3KCA*, *AKT1*, and *TP53* mutations	[57]
Di Leo, A. et al.	2018	*PIK3CA* mutations	[59]
Chin, Y. et al.	2021	57 SNV * the majority were in *TP53* (37%), *PIK3CA* (21%), *AKT1* (7%), *EGFR* (5%) and *KRAS* (5%)	[61]
Yi, Z. et al.	2020	*TP53* mutations	[70]
Visvanathan, K. et al.	2017	Cumulative methylation index (CMI) of a 6-gene panel (*AKR1B1*, *HOXB4*, *RASGRF2*, *RASSF1*, *HIST1H3C* and *TM6SF1*)	[128]
Fackler, M. et al.	2021	9-gene panel of breast-cancer-specific DNA methylation markers	[129]
Gerratana, L. et al.	2020	Methylation status of *ESR1* main promoters (promA and promB)	[131]
Turner, N. et al.	2023	One or two mutations from panel of genes	[132]
Chen, Y. et al.	2017	*TP53*, *PIK3CA*, *CDKN2A* from panel of genes	[133]
Takahashi, H. et al.	2017	Methylation of the promoter region of *RASSF1A*	[134]
Connolly, R. et al.	2018	10-gene panel; cumulative methylation index (CMI)	[135]
Han, Z. et al.	2017	Methylation of *RASSF1A* and *WIF-1*	[136]
Jank, P. et al.	2020	Methylation of 5 CpG islands of *MGMT* promoter	[137]
Lin, P. et al.	2021	Target gene panel (14 genes)	[138]
Liu, B. et al.	2022	*TP53* mutations	[139]
Guan, X. et al.	2020	*HER2* amplification	[140]
Rothé, F. et al.	2019	*PIK3CA* and/or *TP53* mutations	[141]
Li, X. et al.	2020	*ESR1* mutations	[142]
Cristofanilli, M. et al.	2022	*ESR1, PIK3CA*, and *TP53* mutations	[143]
Bidard, F. et al. Berger, F. et al.	2022 2022	*ESR1* mutations	[55,144]
Turner, N. et al.	2020	*PIK3CA*, *ESR1*, *HER2*, *PTEN*, and *AKT1* mutations	[145]
Lyu, D., et al.	2022	Mutations in *PI3K/AKT/mTOR* signaling pathway, *ESR1*, *HER2* mutations	[146]
Mastoraki, S. et al.	2018	*ESR1* methylation	[147]
Chimonidou, M. et al.	2017	Methylation of the promoter region of *SOX17*	[148]
Panagopoulou, M. et al.	2019	Methylation status of a panel of cancer-related genes (*KLK10*, *SOX17*, *WNT5A*, *MSH2*, *GATA3*)	[149]
ctDNA studies using whole-genome sequencing			
Parsons, H. et al.	2020	Exome sequencing to identify patient-specific SNVs, design custom minimal residual disease tests	[18]
Magbanua, M. et al.	2021	Whole-exome sequencing, design of individual mutation panels	[19]
Moss, J. et al.	2020	3 target regions specifically hypermethylated or hypomethylated uniquely in breast cancer from whole-genome bioinformatic analysis	[43]
Widschwendter, M. et al.	2017	Representation bisulfite sequencing (RRBS), found specific region EFC#93 analysis via ultra-high coverage bisulfite sequencing	[48]
McDonald, B. et al.	2019	Exome sequencing of tumor biopsies and analysis of dozens to hundreds of mutations in serial plasma samples	[69]
Garcia-Murillas, I. et al.	2015	Targeted NGS of exons of 273 genes	[150]
Coombes, R. et al.	2019	Ultra-deep sequencing	[151]
Garcia-Murillas, I. et al.	2019	Identify somatic mutations by breast cancer driver gene panel, design of individual mutation panels	[152]
Radovich, M. et al.	2020	Big commercial platform covering multiple genes	[153]
Yi, Z. et al.	2021	Target-capture deep sequencing of 1021 genes, calculation of molecular tumor burden index	[154]

* SNV—single nucleotide variant.

**Table 3 ijms-24-17073-t003:** Correlation of ctDNA with tumor burden (possibility of early detection of recurrence/metastases).

Author, Year [Reference]	Number of Patients	Characteristic of Patients, Trial	Studied Parameter	Method, Tumor/ctDNA	Main Results
Parsons, H., 2020 [18]	158	Metastatic breast cancer ER + HER2- (*n* = 16) and non-metastatic breast cancer 0-III stage (*n* = 142)	Exome sequencing to identify patient-specific SNVs ^2^, custom test	NGS	Whole-exome sequencing of tumors was performed and individualized MRD ^3^ tests were designed. MRD detection at 1 year was strongly associated with distant recurrence (HR = 20.8; 95% CI 7.3–58.9). Median lead time from first positive sample to recurrence was 18.9 months (range = 3.4–39.2 months)
Fackler, M., 2021 [42]	72	Metastatic breast cancer (*n* = 46), benign breast disease (*n* = 17), healthy normal controls (*n* = 9)	9-gene panel of breast-cancer-specific DNA methylation markers	Methylation specific-PCR in automated Liquid Biopsy for Breast Cancer Methylation (LBx-BCM) prototype	9-gene panel of methylated DNA markers that discriminates stage IV BC from benign breast disease and healthy normal subjects using ctDNA was identified. This assay has potential clinical utility in monitoring therapeutic response and predicting disease recurrence.
Widschwendter, M., 2017 [48]	419	Breast cancer, SUCCESS trial	Representation bisulfite sequencing (RRBS) in tissue, ultra-high coverage bisulfite sequencing in serum; specific region EFC#93 (a pattern of five, linked CpGs methylated in BC) analysis	NGS	EFC#93 was an independent poor prognostic marker in pre-chemotherapy samples (HR for death = 7.689) and superior to circulating tumor cells (CTCs) (HR for death = 5.681). More than 70% of patients with both CTCs and EFC#93 serum DNAme positivity in their pre-chemotherapy samples relapsed within five years. EFC#93 positivity after chemotherapy is significantly (*p* = 0.014) less frequently observed in early stage (T1) compared to late stage (T2–4) cancers. EFC#93 serum positivity before chemotherapy was a very strong marker of poor prognosis, for both DFS and OS ^4^.
Riva, F., 2017, [54]	46	Nonmetastatic triple-negative breast cancer	*TP53*	NGS/ddPCR	Correlation with mitotic index (*p* = 0.003), tumor grade (*p* = 0.003), and stage (*p* = 0.03)
Chin, Y., 2021 [61]	109	I–IV stages; 83%—luminal HER2 negative, 6%—HER2 positive, 11%—triple-negative	Oncomine Pan-Cancer Cell-Free Assay panel (52 genes) *TP53*, *PIK3CA*, *AKT1*, *EGFR*, *KRAS*	NGS	Correlation of the frequency of detection of ctDNA with the prevalence of the disease: stage (*p* = 0.00026), involvement of lymph nodes (*p* = 0.00649) and presence of distant metastases (*p* = 0.0005)
Gerratana, L., 2020 [131]	49	Metastatic breast cancer ER + HER2-	*ESR1* epigenetic status was defined by assessing the methylation of its main promoters (promA and promB) in ct DNA	Methylation- specific digital droplet PCR (MS-ddPCR)	No significant impact on PFS was observed for main promoters of *ESR1*: promA (*p* = 0.3777) and promB (*p* = 0.7455) dichotomized at the median while a ≥2-fold increase in promB or in either promA or promB after 3 months hormonotherapy resulted in a significantly worse prognosis (*p* = 0.0189, *p* = 0.0294, respectively). A significant increase after 3 months hormonotherapy was observed for promB among patients with *PIK3CA* mutation (*p* = 0.0173). Significantly lower promA levels at baseline were observed in patients with liver metastases (*p* = 0.0212).
Turner, N., 2023 [132]	208	Triple-negative breast cancer after primary treatment, c-TRAK TN trial	RMH200 gene panel (200 cancer genes), or the ABC-BIO panel (41 gene) for tumor, using one or two mutations in cfDNA	NGS/dPCR	71.9% patients (23/32, 95% CI 53.3–86.3%) had metastatic disease on staging at the time of ctDNA detection. Median lead time between ctDNA detection and disease recurrence in the intervention group was 1.6 months (95% CI 1.2–4.9 months)
Chen, Y., 2017 [133]	38	Triple-negative breast cancer after primary treatment	Oncomine Research Panel consisting of 134 cancer genes (*TP53*, *PIK3CA*, *CDKN2A*)	NGS	ctDNA mutations in the plasma were detected of four patients (three *TP53* mutations, one *AKT1* mutation, one *CDKN2A* mutation). All four patients had recurrence of their disease (100% specificity), but sensitivity was limited to detecting only 4 of 13 patients who clinically relapsed (31% sensitivity). The analysis did not identify any de novo mutations exclusively in the plasma, suggesting that only mutations first identified in the primary tumor were detectable in the plasma Patients with detectable circulating tumor DNA had an inferior DFS (*p* < 0.0001; median DFS: 4.6 mos. vs. not reached; HR = 12.6, 95% CI: 3.06–52.2)
Garcia-Murillas, I., 2015 [150]	55	Early breast cancer	Panel targeting 14 breast cancer driver genes for tumor, mutation-specific dPCR assays for cfDNA	NGS/ddPCR	Detection of ctDNA in patients who received therapy before surgery was predictive of early relapse (HR = 12.0): the median DFS ^1^ was 13.6 months (ctDNA detected) versus median not reached (ctDNA not detected). In total, 50% of the patients who relapsed in the study had ctDNA detected in a single postsurgical sample and 80% had ctDNA detected in serial samples. Of the patients who did not relapse, 96% did not have ctDNA detected in either a single postsurgical sample (*p* = 0.0038) or serial samples (*p* < 0.0001)
Coombes, R., 2019 [151]	49	Breast cancer after primary treatment	Whole genome sequencing of tumor, ultra-deep sequencing of 16 individual somatic variants in cfDNA	NGS	Plasma ctDNA was detected ahead of clinical or radiologic relapse in 16 of the 18 relapsed patients (sensitivity of 89%); metastatic relapse was predicted with a lead time of up to 2 years (median 8.9 months; range 0.5–24.0 months). None of the 31 non-relapsing patients were ctDNA-positive at any time point across 156 plasma samples (specificity of 100%)

^1^ DFS—disease-free survival; ^2^ SNV—single nucleotide variant; ^3^ MRD—minimal residual disease; ^4^ OS—overall survival.

**Table 4 ijms-24-17073-t004:** ctDNA as a surrogate marker for the effectiveness of neoadjuvant drug therapy and minimal residual disease.

Author, Year [Reference]	Number of Patients	Characteristics of Patients, Trial	Studied Parameter	Method, Tumor/ctDNA	Main Results
Magbanua, M., 2021 [19]	84	Early breast cancer	Whole-exome sequencing, design patient-specific custom test	NGS	Patients who remained ctDNA positive at T1 (3 week after initiation of paclitaxel) were significantly more likely to have residual disease after NAC (83% non-pCR) compared with those who cleared ctDNA (52% non-pCR; OR = 4.33, *p* = 0.012). After NAC, all patients who achieved pCR were ctDNA negative (n = 17, 100%). Patients who did not achieve pCR but were ctDNA negative (86%) had excellent outcomes, similar to those who achieved pCR (HR = 1.4; 95% CI 0.15–13.5).
Moss, J., 2020 [43]	29	stage IIA–IIIC	3 target regions specifically hypermethylated or hypomethylated uniquely in breast cancer received via bioinformatic analysis	NGS	Levels of methylation of ctDNA during the last month of NAC could predict the presence of residual disease (*p* = 0.006) and were significantly lower than at the start of treatment for patients with a pCR but not for patients with residual disease (*p* = 0.008 and *p* = 0.58, respectively). The association between methylation of ctDNA and residual disease is strong even when taking into account other factors such as age, receptor status, and stage.
Riva, F., 2017 [54]	46	Nonmetastatic triple-negative breast cancer	*TP53*	NGS	ctDNA positivity after 1 cycle of NAC ^1^ was correlated with shorter DFS ^2^ (*p* < 0.001) and overall (*p* = 0.006) survival.
McDonald, B., 2019 [69]	33	Nonmetastatic breast cancer	Exome sequencing of tumor biopsies and analysis of dozens to hundreds of mutations in serial plasma samples	Targeted digital sequencing (TARDIS)	TARDIS results were informative in 100% of the samples. Patients with pCR ^4^ showed a large decrease in ctDNA concentration during therapy.
Turner, N., 2023 [132]	208	Triple-negative breast cancer after primary treatment, c-TRAK TN trial	RMH200 gene panel (200 cancer genes), or the ABC-BIO panel (41 gene) for tumor, using one or two mutations in cfDNA	NGS/dPCR	71.9% patients (23/32, 95% CI 53.3–86.3%) had metastatic disease on staging at the time of ctDNA detection. Median lead time between ctDNA detection and disease recurrence in the intervention group was 1.6 months (95% CI 1.2–4.9 months). The rapid relapsing nature of high-risk triple-negative BC challenged implementation of MRD ^3^ detection.
Takahashi, H., 2017 [134]	87	Breast cancer II–III stage	Methylated ctDNA (met-ctDNA) for the promoter region of *RASSF1A*	One-step methylation-specific PCR (OS-MSP)	In the patients with positive met-ctDNA before NAC, met-ctDNA significantly decreased after NAC in those with disease that responded to therapy (*p* = 0.006), but not in patients whose disease did not respond to therapy. Met-ctDNA after NAC was found to be significantly (*p* = 0.008) correlated to the extent of residual tumor burden.
Connolly, R., 2018 [135]	62	Breast cancer	10-gene panel; cumulative methylation index (CMI)	Methylation- specific PCR	High tissue CMI levels at 15th day of treatment may predict poor response. Increase in tissue CMI levels at 15th day of treatment was associated with 40% lower chance of obtaining pCR (OR = 0.60, 95% CI 0.37–0.97; *p* = 0.037).
Han, Z., 2017 [136]	126	Advanced breast cancer	Methylation of *RASSF1A* and *WIF-1*	Methylation- specific PCR	Positive rates of *RASSF1A* methylation and *WIF-1* in serum of the patients in the effective group were significantly lower than those in the ineffective group (*p* = 0.002 and *p* = 0.001, respectively), the mRNA of RASSF1A and WIF-mRNA was significantly higher than the ineffective group (*p* < 0.05).
Jank, P., 2020 [137]	174	Triple-negative breast cancer II-III stage, GeparSixto trial	Methylation of 5 CpG islands of *MGMT* promoter	Pyrosequencing	*MGMT* promoter methylation was not significantly associated with pCR rate, and was not related to different chemotherapy response rates in the triple-negative BC.
Lin, P., 2021 [138]	60	Breast cancer II–III stage	Deep sequencing of a target gene panel (14 genes)	NGS	The presence of ctDNA after NAC was a robust marker for predicting relapse in stage II-to-III BC patients (HR = 4.29, 95% CI 2.06–8.92, *p* < 0.0001)
Garcia-Murillas, I., 2015 [150]	55	Early breast cancer	Panel targeting 14 breast cancer driver genes for tumor, mutation-specific dPCR assays for cfDNA	NGS/ddPCR	Detection of ctDNA in patients who received NAC before surgery in serial samples was predictive of early relapse (HR = 12.0): the median disease-free survival was 13.6 months (ctDNA detected) versus median not reached (ctDNA not detected). In total, 50% of the patients who relapsed in the study had ctDNA detected in a single postsurgical sample and 80% had ctDNA detected in serial samples. Of the patients who did not relapse, 96% did not have ctDNA detected in either a single postsurgical sample (*p* = 0.0038) or serial samples (*p* < 0.0001).
Garcia-Murillas, I., 2019 [152]	101	Early breast cancer	Breast cancer driver gene panel, design of individual mutation panels	NGS	ctDNA detection during follow-up was associated with a high rate of relapse.
Radovich, M., 2020 [153]	142	Early triple-negative breast cancer, BRE12-158	Commercial platform covering multiple genes (FoundationACT^®^ or FoundationOneLiquid Assay^®^)	NGS	Detection of ctDNA and circulating tumor cells in triple-negative BC patients after NAC was associated with disease recurrence.

^1^ NAC—neoadjuvant chemotherapy; ^2^ DFS—disease-free survival; ^3^ MRD—minimal residual disease; ^4^ pCR—pathological complete response.

**Table 5 ijms-24-17073-t005:** ctDNA mutations as prognostic and predictive factors for the effectiveness of therapy.

Author, Year [Reference]	Number of Patients	Characteristics of Patients, Trial	Studied Parameter	Method	Main Results
Fribbens, C., 2016 [20]	161 + 360	Metastatic ER+ breast cancer, SoFEA trial, PALOMA3 trial	*ESR1* mutation	ddPCR	SoFEA: *ESR1* mutations were detected in the ctDNA of 39.1% of the patients (63 of 161). *ESR1* mutation in their ctDNA had worse PFS than those with the *ESR1* wild type when treated with aromatase inhibitor exemestane (2.6 months versus 8.0 months, HR = 2.12; *p* = 0.01), but not fulvestrant. In *ESR1* mutations, fulvestrant resulted in higher PFS compared to exemestane (HR 0.52; 95% CI 0.30–0.92; *p* = 0.02). PALOMA3: *ESR1* mutations were detected in the ctDNA of 25.3% of the patients (91 of 360). Presence of *ESR1* mutations in ctDNA of advanced BC patients showed worse PFS compared with those with the *ESR1* wild type (HR = 1.46, *p* = 0.02).
Bidard, F., 2022 [55]	1017	Metastatic ER+ HER2- breast cancer, PADA-1 trial	*ESR1*	ddPCR	Earlier detection of *ESR1* mutation growth as a marker of progression and early (before the appearance of traditional clinical and radiological signs) change in therapy ensured a gain in PFS: 11.9 and 5.7 months. (HR = 0.61, 95% CI 0.43–0.86; *p* = 0.004).
Tolaney, S., 2022 [56]	669	Advanced ER+ HER2- breast cancer, MONARCH 2 trial	Mutations *PIK3CA*, *ESR1*	ddPCR	Increase in median PFS with the addition of abemaciclib to fulvestrant (vs. placebo + fulvestrant) in both wild-type *PIK3CA* (median 16.9 vs. 12.3 months; HR = 0.51, 95% CI 0.33–0.78) and *PIK3CA* mutation (median 17.1 vs. 5.7 months, HR = 0.53; 95% CI 0.33–0.84); as well as with wild-type *ESR1* (median 15.3 vs. 11.2 months, HR = 0.44, 95% CI 0.27–0.71) and with *ESR1* mutation (median 20.7 vs. 13.1 months; HR 0.54; 95% CI 0.37–0.79).
Sabatier, R., 2022 [57]		HER2-negative advanced breast cancer, TAKTIC trial	Low-coverage whole-genome sequencing for all plasma samples; ddPCR for some patients with driver mutations of *PI3KCA*, *AKT1*, and *TP53* in their tumors	NGS + ddPCR	The presence of ctDNA upon inclusion was correlated with PFS (6-month PFS was 92% for ctDNA-negative patients versus 68% for ctDNA-positive cases (HR = 3.45, *p* = 0.007)). Copy number alterations were associated with disease progression under paclitaxel-LY2780301. Therefore, ctDNA detection at baseline was associated with shorter PFS, while plasma-based copy number analysis helped to identify alterations involved in resistance to AKT/p70S6K inhibitor plus paclitaxel treatment.
Di Leo, A., 2018 [59]	348	Locally advanced and metastatic ER+ HER2- breast cancer, BELLE-3 trial	*PIK3CA* mutation	Inostics BEAMing assay	Median PFS was significantly longer in the buparlisib versus placebo group (3.9 months vs. 1.8 month (HR = 0.67, 95% CI 0.53–0.84, *p* = 0.0003) in patients with PIK3CA mutations detected in tumor tissue or ctDNA isolated upon study entry.
Yi, Z., 2020 [70]	804	Metastatic breast cancer	*TP53*	NGS	*TP53* mutations were associated with a shorter DFS vs. wild-type *TP53* (HR = 1.32, 95% CI = 1.09–1.61, *p* = 0.005); *TP53* mutations in exons 5–8 were associated with worse outcome (HR = 1.50, 95% CI = 1.11–2.03, *p* = 0.009); *TP53* mutation status was not significantly associated with PFS in HER2-positive patients who received first-line trastuzumab-based therapy (*p* = 0.966). In the taxane combination group, patients with *TP53* mutations exhibited longer PFS than those without *TP53* mutations (HR = 0.08, 95% CI = 0.02–0.30, *p* < 0.001). In the non-taxane combination group, patients with *TP53* mutations displayed shorter PFS than those with wild-type *TP53* (HR = 4.84, *p* = 0.005).
Visvanathan, K., 2017 [128]	141	Metastatic breast cancer	Cumulative methylation index (CMI) of a minimal 6-gene subset (*AKR1B1*, *HOXB4*, *RASGRF2*, *RASSF1*, *HIST1H3C* and *TM6SF1*)	Quantitative multiplex assay based on multiplex nested real-time PCR (cMethDNA)	Median PFS and OS were significantly shorter in women with a high CMI (PFS = 2.1 months; OS = 12.3 months) versus a low CMI (PFS = 5.8 months; OS = 21.7 months). In multivariable models, among women with metastatic BC, a high versus low CMI at week 4 was independently associated with worse PFS (HR = 1.79; 95% CI 1.23–2.60; *p* = 0.002) and OS (HR = 1.75; 95% CI 1.21–2.54; *p* = 0.003).
Chen, Y., 2017 [133]	38	Triple-negative breast cancer after primary treatment	Oncomine Research Panel consisting of 134 cancer genes for tumor; *TP53*, *AKT*, *CDKN2A* in cfDNA	NGS	Patients with detectable ctDNA had an inferior DFS (*p* < 0.0001; median DFS: 4.6 mos. vs. not reached; HR = 12.6, 95% CI: 3.06–52.2).
Liu, B., 2022 [139]	1184	HER2-positive breast cancer, Geneplus cohort	*TP53*	NGS	*TP53* mutations were associated with a shorter PFS ^1^ (*p* = 0.004) on anti-HER2 antibody therapy; the value of *TP53* mutation in predicting HER2 tyrosine kinase inhibitor response was inconsistent.
Guan, X., 2020 [140]	105	HER2-positive breast cancer	*HER2* copy numbers	NGS	Correlation of the number of copies of the *HER2* gene before treatment with the frequency of objective effects (*p* = 0.010); consistently high copy number after 6 weeks was associated with a decrease in DFS ^2^ (*p* < 0.001).
Rothé, F., 2019 [141]	69	Early HER2+ breast cancer, NeoALTTO trial	*PIK3CA* and/or *TP53* mutations	ddPCR	ctDNA detection before neoadjuvant anti-HER2 therapy was associated with low pCR ^3^ rates. Patients with HER2-enriched tumors and undetectable ctDNA at baseline had the highest pCR rates.
Li, X., 2020 [142]	45	Metastatic ER+ breast cancer	Targeted NGS panel of 425 genes; *TP53* mutation; *ESR1* mutation	NGS	Six genes: *TP53* (64.4%), *PIK3CA* (46.7%), *ESR1* (20%), *ERBB2* (15.6%), *ATM* (15.6%), and *BRCA1* (13.3%), were mutated in more than 13% of the patients. Patients with *TP53* mutations (29 patients of 45) had significantly worse OS ^4^ than the carriers of wild-type alleles (*p* = 0.0094). *ESR1* mutations were recurrently enriched in ER+ metastatic BC patients but were rarely present in primary tumor tissues. The median time from aromatase inhibitor endocrine therapy to the first detection of *ESR1* mutations was 39 months (95% CI 21.3–57.6). The change in allele frequency of *ESR1* mutations (observed in 9 of 45 patients) was an important biomarker, which could predict endocrine resistance of ER+ BC. Therapy with everolimus in four cases with acquired *ESR1* mutations showed longer PFS.
Cristofanilli, M., 2022 [143]	331	Metastatic ER+ HER2- breast cancer, PALOMA3 trial	Panel of 17 driver and CDK4/6-related genes; analysis of *ESR1*, *PIK3CA*, *TP53* mutations	NGS	Favorable OS in the palbociclib (+fulvestrant) vs. placebo (+fulvestrant) group was observed regardless of *ESR1*, *PIK3CA*, or *TP53* mutation status; *ESR1*, *PIK3CA* and or *TP53* mutations were prognostic for OS (HR = 1.58, 1.44 and 2.19, consequently).
Turner, N., 2020 [145]	1034	Advanced breast cancer, PlasmaMATCH trial	Mutations *PIK3CA*, *ESR1*, *HER2*, *PTEN*, and *AKT1*	dPCR + NGS	Neratinib for *HER2*-mutant BC and capivasertib for *AKT1*-mutant BC identified by ctDNA testing had comparable activity to that observed when guided by tissue testing in previous study, respectively. ctDNA testing enables the selection of mutation-directed therapies for patients with BC.
Lyu, D., 2022 [146]	113	Metastatic ER+ breast cancer	Whole genome sequencing, *PI3K/AKT/mTOR* signaling pathway, *ESR1*, *HER2* mutations	NGS	The risk of progression was lower in groups of patients who received treatment in accordance with the mutational status (HR = 0.55, *p* = 0.023)
Mastoraki, S., 2018 [147]	58	ER+ HER2- metastatic breast cancer	*ESR1* methylation in CTCs and paired plasma ctDNA	Methylation- specific PCR	*ESR1* methylation in CTCs and a high concordance with paired plasma ctDNA were reported. In serial peripheral blood samples of patients treated with everolimus/exemestane, *ESR1* methylation was observed in 10/36 (27.8%) CTC-positive samples, and was associated with lack of response to treatment (*p* = 0.023).
Chimonidou, M., 2017 [148]	153	Breast cancer patients and healthy individuals	DNA methylation status of *SOX17*, *CST6* and *BRMS1* promoters in CTCs and ctDNA	Methylation- specific PCR	Association between the EpCAM (epithelial cell adhesion molecule)-positive CTC-fraction and ctDNA for *SOX17* promoter methylation both for patients with early (*p* = 0.001) and metastatic BC (*p* = 0.046) was reported but not for *CST6* and *BRMS1*. In early BC, *SOX17* promoter methylation in the EpCAM-positive CTC-fraction was associated with CK-19 mRNA expression (*p* = 0.006) and worse OS (*p* = 0.044). In the metastatic setting, *SOX17* promoter methylation in ctDNA was highly correlated with CK-19 (*p* = 0.04) and worse OS (*p* = 0.016).
Panagopoulou, M., 2019 [149]	235	150 and 16 breast cancer patients under adjuvant and neoadjuvant therapy, respectively, 34 patients with metastatic disease and 35 healthy volunteers	Methylation status of a panel of cancer-related genes (*KLK10*, *SOX17*, *WNT5A*, *MSH2*, *GATA3*)	Methylation- specific PCR	Methylation of at least 3 or 4 genes was significantly correlated to shorter OS and no pharmacotherapy response, respectively. Classification analysis by a fully automated, machine learning software (JADBio software, Gnosis Data Analysis) produced a single-parametric linear model using cfDNA plasma concentration values, with great discriminating power to predict response to therapy (AUC 0.803, 95% CI 0.606–1.000) in the metastatic group. Two more multi-parametric signatures were produced for the metastatic group, predicting survival and disease outcome. A multiple logistic regression model was constructed, discriminating between patient groups and healthy individuals. cfDNA emerged as a highly potent predictive classifier in metastatic BC.
Yi, Z., 2021 [154]	125	Metastatic breast cancer, CAMELLIA trial	Target-capture deep sequencing of 1021 genes to detect somatic variants in ctDNA; determining the molecular tumor burden index (mTBI)	NGS	High-level pretreatment mTBI was correlated with shorter OS (*p* = 0.011). Patients with an mTBI decrease to less than 0.02% at the first tumor evaluation had longer PFS and OS (*p* < 0.001 and *p* = 0.007, respectively). The patients classified as molecular responders had longer PFS and OS than the nonmolecular responders (*p* < 0.001 and *p* = 0.036, respectively).
Murtaza, M., 2015 [172]	1	Metastatic ER+ HER2+ breast cancer	Genomic architecture and infer clonal evolution in eight tumor biopsies and nine plasma samples collected over 1193 days of clinical follow-up were characterized using exome and targeted amplicon sequencing	NGS	Ubiquitous stem mutations (common to all tumor biopsies) have the highest circulating levels in plasma followed by metastatic-clade and private mutations. In addition, serial changes during treatment in circulating levels of private somatic mutations correlate with disease progression in their respective tumor lesions on imaging.

^1^ PFS—progression-free survival; ^2^ DFS—disease-free survival; ^3^ pCR—pathological complete response; ^4^ OS—overall survival.

## Data Availability

Data sharing not applicable. No new data were created or analyzed in this study. Data sharing is not applicable to this article.

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
