# Peer review of "Circulating Tumor DNA Is a Variant of Liquid Biopsy with Predictive and Prognostic Clinical Value in Breast Cancer Patients"

_ijms, 2023, doi:10.3390/ijms242317073_

Round 1

Reviewer 1 Report (New Reviewer)

Comments and Suggestions for Authors

This review provides a very complete and comprehensive review on the ctDNA detection for Breast Cancer Diagnosis and Prognosis.  Starting from the methodologies to clinical applications, I feel it is well organized and presented.  Several useful tables have also been arranged to summarize the progression for ctDNA clinical applications, which will be useful for the readers to follow up.  One suggestion is regarding the combined diagnosis/prognosis methods for ctDNA with other biomarkers (such as CTCs or CTM...) , may be considered to add to increase the impact of this review to more related fields for future perspectives. 

Author Response

Thank you very much for taking the time to review our manuscript. We have added information about the possibility of a combined approach for diagnostic and prognostic markers based on ctDNA and СТС and CTM (lines  208-210), but we considered that a detailed analysis of these approaches would make the review very redundant.

Reviewer 2 Report (New Reviewer)

Comments and Suggestions for Authors

A very well-written review that discusses a wide range of cfDNA topics, emphasizing its application in breast cancer. The authors provide a detailed historical perspective that helps situate the topic within its development, allowing the readers to understand how the field has evolved, the key milestones, and the progression of ideas. This context is crucial for readers to grasp the significance and relevance of the current state of research in the field. The authors also briefly discuss the concept of cell-bound DNA. However, we remain keen on this subject and its relevance in the ctDNA analysis for monitoring breast cancer patients.

The section (section 2.4) on how to study cfDNA is interesting. This is an extremely hot topic in cfDNA research. Although the authors discuss some means to study cfDNA using NGS, it would have been interesting to include a comment on the emerging use of the Nanopore technology. Although this technology platform has some limits in detecting specific mutations, it provides several advantages to studying DNA methylation profiles without the need for chemical processing of DNA (via bisulphite). It also provides a rapid means to study non-CpG DNA methylation in the nuclear and/or mitochondrial ctDNA, another increasingly studied (and perhaps controversial) area of research. In fact, the lack of discussion on the importance of DNA methylation is the major weakness of this review. DNA methylation of circulating cell-free tumor DNA (ctDNA) is an important area of research in cancer, including breast cancer. DNA methylation markers in ctDNA can be used for early detection and diagnosis of breast cancer. Methylation changes in specific genes or regions associated with breast cancer can be detected in the blood, even during the early stages of the disease. This offers the potential for non-invasive screening approaches that may complement or enhance existing diagnostic methods.

But otherwise, I enjoyed reading the review. Reviews on the use of ctDNA is a crowded space. However, this review has some strengths and should be published, albeit with the addition of a section on the importance of DNA methylation when using ctDNA as a clinical tool.

Author Response

Thank you very much for your time and helpful comments. The epigenetic direction of research has not yet been so developed for ctDNA as mutation analysis. The main works carried out recently regarding epigenetic changes in ctDNA and having clinical significance have been reflected in revised review. The topic of ctDNA methylation in cancer and breast cancer is discussed in section 2.4. “Possible applications of ctDNA in oncology” and 3.1. “Application of ctDNA for early detection/screening of breast cancer”. We further emphasized the importance of ctDNA methylation for the diagnosis and early detection of breast cancer (lines 365-367, 408-413, 422-425). In section 3.2 “ctDNA as a predictive and prognostic marker in breast cancer treatment”, we added studies investigating the use of ctDNA methylation for treatment response (lines 466-468, 496-499, 531-533, 591-600, 610-612, 740-743). We also added methodological approaches to ctDNA analysis, and a discussion on the advantages of the Nanopore technology to studying DNA methylation profiles in cancer patients (lines 248-252, 265, 311-313, 326-333).

This manuscript is a resubmission of an earlier submission. The following is a list of the peer review reports and author responses from that submission.

Round 1

Reviewer 1 Report

Comments and Suggestions for Authors

The review "Circulating tumor DNA is a variant of liquid biopsy with predictive and prognostic clinical value in breast cancer patients" is a study of an important medical issue, which is breast cancer.                                 This review is based on results already published in the scientific literature. It is not a revealing study, as evidenced by some of the references cited by the Authors published 20 years ago.                                                                  Many parts of this review are borrowed from  the other articles and reviews.

Comments on the Quality of English Language

Moderate English editing and proofreading by an English-language translator is required.

Author Response

Thank you for your time and comments. By reading your comments, it is clear that you are closely following this research field, but this is likely not the case of all clinicians. Our aim was to collect in one review information that allows the general reader (both clinicians and biologists) to be consistently introduced to the field of liquid biopsy, focusing on the work on ctDNA in breast cancer. Therefore, we felt it necessary to include early works that are of historical interest or provide important information on the topic, and in some cases it was important for us to refer to the original work, and not to their citation by other later authors. This review is based on clinical trials conducted over the past few years on the use of ctDNA for the treatment of BC, as well as important earlier works.

Since the issues of liquid biopsy and ctDNA analysis for various applications are widely discussed in the scientific literature, a detailed description of the results obtained in the works on ctDNA research in breast cancer can help the reader in an accurate understanding of the current state of the problem and the details of the studies already carried out. We have tried to summarize and rethink the sections devoted to introducing the reader to the field of liquid biopsy and the place of ctDNA in it.

Thank you very much for your review.

Reviewer 2 Report

Comments and Suggestions for Authors

The manuscript by Zavarikyna et al.: "Circulating tumor DNA is a variant of liquid biopsy with predictive and prognostic clinical value in breast cancer patients" promises a focused summary of the use of ctDNA in breast cancer. Hence the reader expects such focus throughout the manuscript. Overall, the manuscript contains important information and is an interesting read. The authors collected much information about the history of nucleic acids in liquid biopsies, background and detection methods, and the potential clinical application. 

Critical comments:

1. The manuscript needs better structuring. If the focus is breast cancer, the authors should try to focus on breast cancer and introduce breast cancer better to the general reader (mutation backgrounds, treatments and clinical outcome).

2. Known ctDNA mutations, methylation etc associated with a breast cancer diagnosis and treatment monitoring should be emphasized more and separately from other cancers mentioned in the manuscript. 

3. Summary of ctDNA detected in other cancers could be summarized under a separate subheading.

4. More tables should be introduced to allow a better overview of ctDNA in clinical application in BC

5. The authors ought to write more in their conclusion about the future application of ctDNA in personalized medicine in breast cancer and how this technique can be applied in tracing therapy response

Minor:

1. Sentences like (lines 66-68): "In addition to ctDNA, the concept of liquid biopsy includes the analysis of circulating tumor cells, circulating tumor RNAs, long non-coding RNAs, messenger RNAs, microRNAs, extracellular vesices....." should be re-written. The words "extracellular vesicles" should be explained. I assume extracellular vesicles are in the above sentence as they also carry DNA, mRNA, miRNA, etc, but need to be explained.

2. Abbreviations like NAC should be written in full at their first use.

3. The quality of the only figure should be improved.

Comments on the Quality of English Language

A native English speaker should read and correct the manuscript.

Author Response

Thank you for your time and precise comments. We have tried to follow all the critical comments in revised manuscript.

  1. We restructured the manuscript focusing on breast cancer and adding the introduction of it to the general reader.
  2. We emphasized information about ctDNA in breast cancer more and separately from other cancers.
  3. Information about ctDNA in other cancers has been placed in a separate subheading.
  4. The section on ctDNA in breast cancer provides additional tables on the various clinical uses of ctDNA in breast cancer treatment.
  5. We supplemented section 3 with a paragraph describing the possibilities of using ctDNA in personalized medicine in breast cancer in the future, so as not to make the conclusion redundant.

The minor critical comments have also been fixed. Thank you very much for your important comments.

Reviewer 3 Report

Comments and Suggestions for Authors

The authors gave an overview of current application of circulating tumor DNA in the prediction and prognosis of breast cancer. The review covered many aspects, including characteristics of ctDNA, methods for detecting ctDNA, and clinical values of ctDNA as a biomarker in BC patients. Here are several points regarding to revision.

1. page 7 line 298, Do the authors mean “pathological complete response” by “pCR”? It’s not clear in the text. 

2. In Table 2, column of “Analysis”, the format or alignment of texts need to be adjusted to make it easier to read. 

3. Section 3.2.1 contains a lot of information and paragraphs, describing from study to study. Re-organizing/ Summarizing the main points is necessary, same with 3.2.2. 

Author Response

Thank you for your time and efforts. We have tried to follow all the points in revised manuscript.

  1. We have checked that all abbreviations are decoded in the text.
  2. Changed the table format for a better perception of the content.
  3. Section 3.2.1 has been reorganized, summarisings have been added. We also made an additional point with a discussion of the clinical significance of the results obtained, based on the work described in paragraph .3.2.1. and 3.2.2. Since there are a large number of works on the topic of ctDNA in the scientific literature, a detailed description of the results obtained in the works on the study of ctDNA in breast cancer can help the reader in an accurate understanding of the details of the studies already carried out and the current state of the problem.

Thank you very much for your helpful and expert comments.

Round 2

Reviewer 3 Report

Comments and Suggestions for Authors

The Authors have addressed all of my concerns with the original manuscript. The paper will make a good contribution to the field.